# Impact of Ocean Currents on Wind Stress in the Tropical Indian Ocean

**Neethu Chacko** [1] , **Meer M. Ali** [2,3,*] **and Mark A. Bourassa** [2,4]

1 Regional Remote Sensing Centre-East, National Remote Sensing Centre, ISRO, Kolkata 700156, West Bengal, India; neethu_c@nrsc.gov.in
2 Centre for Ocean-Atmospheric Prediction Studies, Florida State University, Tallahassee, FL 32310, USA; bourassa@coaps.fsu.edu
3 Andhra Pradesh State Disaster Management Authority, Kunchanapalli 522501, Andhra Pradesh, India
4 Department of Earth, Ocean and Atmospheric Science, Florida State University, Tallahassee, FL 32310, USA
* Correspondence: mmali@coaps.fsu.edu

**Abstract:** This study examines the effect of surface currents on the bulk algorithm calculation of wind stress estimated using the scatterometer data during 2007–2020 in the Indian Ocean. In the study region as a whole, the wind stress decreased by 5.4% by including currents in the wind stress equation. The most significant reduction in the wind stress is found along the most energetic regions with strong currents such as Somali Current, Equatorial Jets, and Agulhas retroflection. The highest reduction of 11.5% is observed along the equator where the Equatorial Jets prevail. A sensitivity analysis has been carried out for the study region and for different seasons to assess the relative impact of winds and currents in the estimation of wind stress by changing the winds while keeping the currents constants and vice versa. The inclusion of currents decreased the wind stress (consistent with scatterometer winds) and this decrease is prominent when the currents are stronger. This study showed that the equatorial Indian Ocean is the most sensitive region where the current can impact wind stress estimation. The results showed that uncertainties in the wind stress estimations are quite large at regional levels and hence better representation of wind stress incorporating ocean currents should be considered in the ocean/climatic models for accurate air-sea interaction studies that are not based on remotely sensed winds.

**Keywords:** scatterometer; wind stress; surface currents; current coupling; Indian Ocean

## 1. Introduction

Winds play an instrumental role in driving the surface currents and also in the air–sea interaction processes. Accurate measurements of wind stress are required to understand the air–sea interaction and other climate variability. Most of the air–sea interaction processes are determined using wind stress, which is a measure of transfer of momentum due to the relative motion between the ocean and atmosphere. Wind stress also exerts surface oceanic circulation, which in turn results in the redistribution of heat and other properties. Wind stress is calculated according to the bulk-aerodynamic formula as,

$$\tau = \rho_a C_d \, U_w{}^2 \tag{1}$$

where, $\rho_a$ is the density of air (1.225 kg/m$^3$), $C_d$ is the dimensionless drag coefficient (approximately $1.3 \times 10^{-3}$, [1]), and $U_w$ the wind velocity. This definition works very well for scatterometer winds, which are calibrated to be relative to the ocean surface [2,3] and has the further advantage of being tuned to produce the correct stress when a neutral drag coefficient is used [4,5]. For ship and buoy winds, the surface current should be vector subtracted from the wind and the stability adjustment is usually non-neutral [6], which is analogous to a change in wind speed between −0.5 to 0.5 ms$^{-1}$. The relative motion

between the atmospheric winds and ocean currents modulate the amount of backscatter (as desired for stress), even though it is primarily determined by the magnitude of winds [7]. Thus, most documentation on scatterometer winds mistakenly refers to the product as a wind rather than the much wordier correct definition of a surface relative equivalent neutral wind [6,8,9]. The wind speeds measured by scatterometers are actually higher or lower than the Earth-relative wind depending on the relative direction of the winds with regards to the ocean currents.

Ref. [2] reported that in the Pacific Ocean, the differences between the scatterometer winds and anemometer winds measured by buoy moorings are explained by the surface currents measured by the buoys. As the ocean currents clearly impact the satellite wind measurements, the effect due to currents should be accounted for while computing the air-sea coupling processes from non-remotely sensed winds. Hence, the relative motion between the ocean and atmosphere should be included in computing the momentum fluxes, in particular the wind stress. Earlier studies have pointed out the importance of including relative motion in the assessment of air-sea interactions. [10] showed that when the effects of currents are included in the simulation of the tropical Atlantic, the equatorial currents are reduced by 30%, which also had an impact on the upwelling and sea surface temperature variability. A basin-wide reduction of wind stress by ~15% is reported by [11] in the northern Pacific Ocean by incorporating currents in the wind stress computation. A similar study by [12] found that improved estimates of sea surface temperature are obtained when the wind stress is computed with relative motion between the ocean and the atmosphere. [13] emphasized the usage of stress-equivalent surface wind speeds from scatterometers than using direct neutral winds by incorporating dependence of stress on air density. To improve computational speed, many models assume that the atmosphere is neutrally stable and that surface currents have no impact on stress.

The inclusion of surface currents into the bulk formula for wind stress modifies Equation (1) to

$$\tau = \rho_a C_d \left| \vec{U}_w - \vec{U}_o \right|^2 \qquad (2)$$

where, $\vec{U}_w - \vec{U}_o$ is the difference between the surface wind ($\vec{U}_w$) and ocean current ($\vec{U}_o$) vectors [11,14,15]. The vectors $\vec{U}_w$ *and* $\vec{U}_o$ were computed from u and v components of both wind and current. Often the inclusion of surface currents in the wind stress is neglected because wind speeds are much higher than surface currents. However, in the oceanic regions where surface currents are stronger and winds are weaker, the perentage impact of it on the wind stress estimates can be higher. This current–wind interaction, which is termed "relative wind stress" [16] is capable of modulating mesoscale processes, vertical upwelling, and momentum transfer [15,17]. Ref. [18] showed that the currents through the modification of wind stress significantly reduce wind power input into the ocean. Ref. [14] examined the impact of current–wind interaction on the upper ocean stratifications and geostrophic circulation in the Bay of Bengal using a regional coupled model simulation. They found that the current inclusion in the wind stress estimation significantly modulated the geostrophic current field and increased the ocean stratification. The impact of currents on air–sea momentum and heat flux exchanges in the Gulf Stream is reported by [15], using a numerical model. The impact of sea surface temperature on the scatterometer-derived wind speed and wind stress is assessed by [19]. Ref. [20] analyzed QuikSCAT Satellite measurements and observed persistent small-scale features in the curl and divergence in ocean wind stress. They also reported clear curl field patterns in the Gulf Stream. This study, however, emphasizes the impact of currents alone on the estimation of satellite-derived wind stress in the Indian Ocean.

Indian Ocean surface currents are unique because of their seasonally reversing nature in response to the reversing monsoonal winds [21–23]. The major currents in the Indian Ocean are the Southwestern Monsoon Current, the Northeastern Monsoon Current, the Somali Current, the East India Coastal Current, the West India Coastal Current, and the

Equatorial Jets. The Summer Monsoon Current flows eastward during the summer season (June–September) and the Winter Monsoon Current flows westward during the winter monsoon (December–February). The Somali currents are very strong western boundary currents that flow northeastwards during the southwest monsoon and southwestward during the northeast monsoon [24]. Ref. [25] reported that the Somali Current can reach up to 2–3 m/s in the summer monsoon season. The Equatorial Jets, however, flow eastward along the equator during the transition months (April–May and October–November) [26]. Thus, studying the impact of currents on wind stress estimation is apt.

In this context, we try to examine the impact of currents on wind stress estimation in the tropical Indian Ocean. Previous studies assessed the impact of currents on wind stress using model simulations [11,27,28]. Ref. [14] using regional coupled model simulations tried to assess the relative wind effect in the Bay of Bengal. However, no study examined the current impact on the wind stress over the different current systems in the Indian Ocean using scatterometer observations. Here, we try to assess the impact of currents on wind stress estimations in the tropical Indian Ocean using scatterometer observations and ocean currents largely estimated from remote sensing. We compare the wind stress estimations through the bulk formula for wind stress with and without the surface currents and demonstrate the direct impact of currents on wind stress in the major current systems of the Indian Ocean.

## 2. Materials and Methods

### 2.1. Data

In this study, we utilize scatterometer winds and satellite-derived surface currents to assess the impact of currents on wind stress estimation in the tropical Indian Ocean. Wind speeds from the Advanced Scatterometer (ASCAT) on-board Metop-A and Metop-B of the European Organisation for the Exploitation of Meteorological Satellites (EUMET-SAT) are utilized in this work. The study region includes the tropical Indian Ocean bounded by 30° E–120° E longitudes and 30° S–30° N latitudes. The daily gridded AS-CAT winds [29] with $0.25° \times 0.25°$ resolution for the period 2007 to 2020 are provided by http://apdrc.soest.hawaii.edu/datadoc/ascat.php (accessed on 3 September 2021). Near-surface currents (averaged over the top 30 m) estimated by Ocean Surface Current Analysis Real-time (OSCAR, [30]) during the same period are used in the study. OSCAR currents are available with $1/3° \times 1/3°$ resolution with 5 days temporal resolution and are obtained from http://apdrc.soest.hawaii.edu/las/v6/dataset?catitem=2845 (accessed on 3 September 2021). The OSCAR product is a direct computation of global surface currents using satellite sea surface height, wind, and temperature. OSCAR currents are calculated using a quasi-steady geostrophic model together with an eddy-viscosity-based wind-driven ageostrophic component and a thermal wind adjustment. The model calculates a surface current averaged over the top 30 m of the upper ocean. The ASCAT winds are re-gridded to match the spatial and temporal resolution of OSCAR surface currents before computing the wind stress. For this, the ASCAT wind magnitudes are first averaged over 5 days temporally and 0.33 degrees spatially. Once the ASCAT winds are brought to the same resolution as OSCAR currents, wind stresses are computed using Equations (1) and (2), where the wind in (1) is the vector sum of scatterometer and OSCAR winds, and the winds in (2) are the scatterometer winds. The stresses computed are then time-averaged to monthly, seasonal, and annual means for further analysis.

### 2.2. Methods

To quantify the effect of currents on the wind stress in the Indian Ocean, this parameter is computed with and without the currents in the wind stress equations. The wind stress computed as described above is referred to as $\tau_{\text{no-Cur}}$ and $\tau_{\text{Cur}}$ estimations, respectively. To assess the relative difference between the strengths of winds and currents in inducing the differences in wind stress, two sets of experiments are conducted by varying the currents and wind speeds. In the first experiment, the wind speeds are increased by keeping the

current speeds as such (EXP1). Wind stress is estimated with and without currents by increasing the wind speeds by 5% and 10%. These experiments are referred as EXP1_W5 and EXP1_W10, respectively. In the second experiment, the current speeds are altered without changing the wind speeds (EXP2). Wind stress is estimated by increasing current speeds by 5% and 10%. These experiments are referred as EXP2_C5 and EXP2_C10, respectively. The estimation without altering either the winds or the currents is referred to as NOEXP. In all the sets of experiments, the $\tau_{no-Cur}$ and $\tau_{Cur}$ are computed and the difference between the two are analyzed.

## 3. Results and Discussions

### 3.1. Annual Impact

The annual means of the wind stress with ($\tau_{Cur}$) and without ($\tau_{no-Cur}$) currents in the Indian Ocean and the difference between the two are presented in Figure 1. The striking feature in the wind stress pattern in both the estimations is over the south-east trade winds, which occur south of 10° S and persist throughout the year in the Indian Ocean [23]. An overall basin-wide reduction in the wind stress can be seen by including currents into the wind stress equation (Figure 1c).

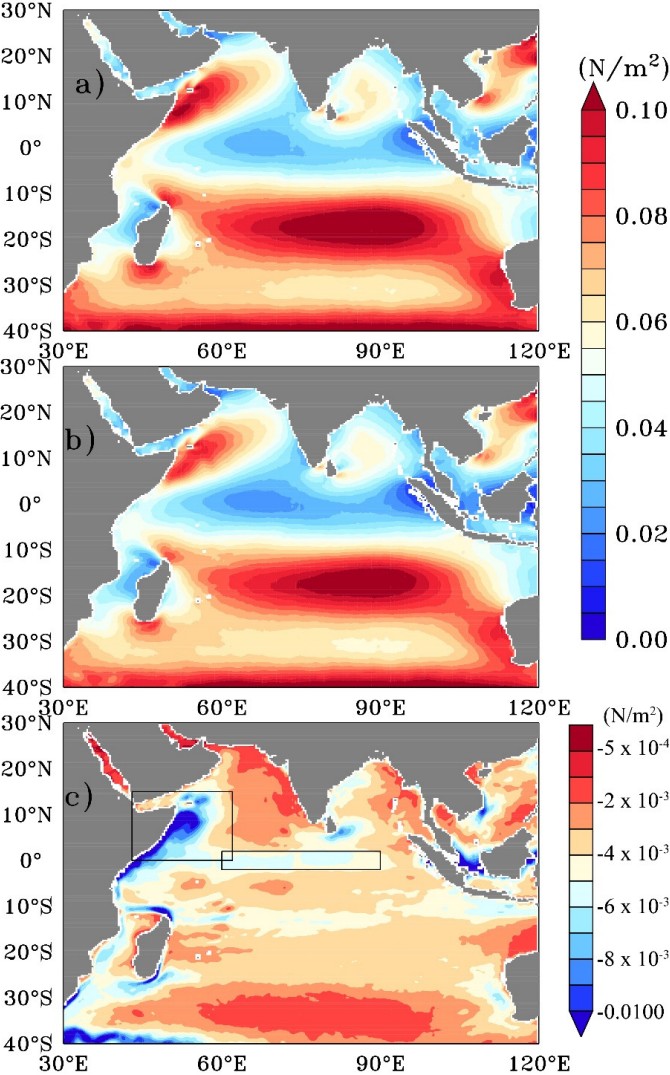

**Figure 1.** Wind stress (N/m$^2$) estimated (**a**) without currents, (**b**) with currents, and (**c**) difference between the two wind stresses (**a**,**b**). The regions off Somali Coast and equatorial Indian Ocean are shown as boxes in (**c**).

Besides a few small-scale features being clearly visible, the notable differences between the two fields ($\tau_{\text{no-Cur}}$ and $\tau_{\text{Cur}}$) are (i) along the region off Somali coast between 0 and 15° N latitudes to the east of 60° E longitude, (ii) a narrow band of region between 30° E–60° E and south of 35° S, which is the regime of Agulhas current, (iii) a band of ±4 degrees of the equator between 60° E and 90° E, and (iv) the region off the eastern coast of India. The difference is negative in the entire study region, indicating that stress without currents is larger than the stress with current. However, the stress differences are found to be negligible in the Arabian Sea and in the Southern Ocean between 40° S and 30° S latitudes.

### 3.2. Zonal and Regional Impact

The zonal average of the wind stress difference (Figure 2) shows that the highest deviation occurs along the equatorial Indian Ocean around 2° N. This large difference at 2° N is because of the large stress difference along 2° N as shown in Figure 1c. Since the currents are stronger in this region [22,23], the difference is also larger. Table 1 summarizes the differences between $\tau_{\text{no-Cur}}$ and $\tau_{\text{Cur}}$ fields for the whole study region as well as for the regions of Somali current and Equatorial Jets. It is noted that by including surface currents into the wind stress computation, the basin-wise averaged wind stress decreased by 5.8%. Notable differences in the percentage reduction of wind stress exist when surface currents are accounted for Somali current region (−9.56%) and Equatorial Jet region (−15.93%).

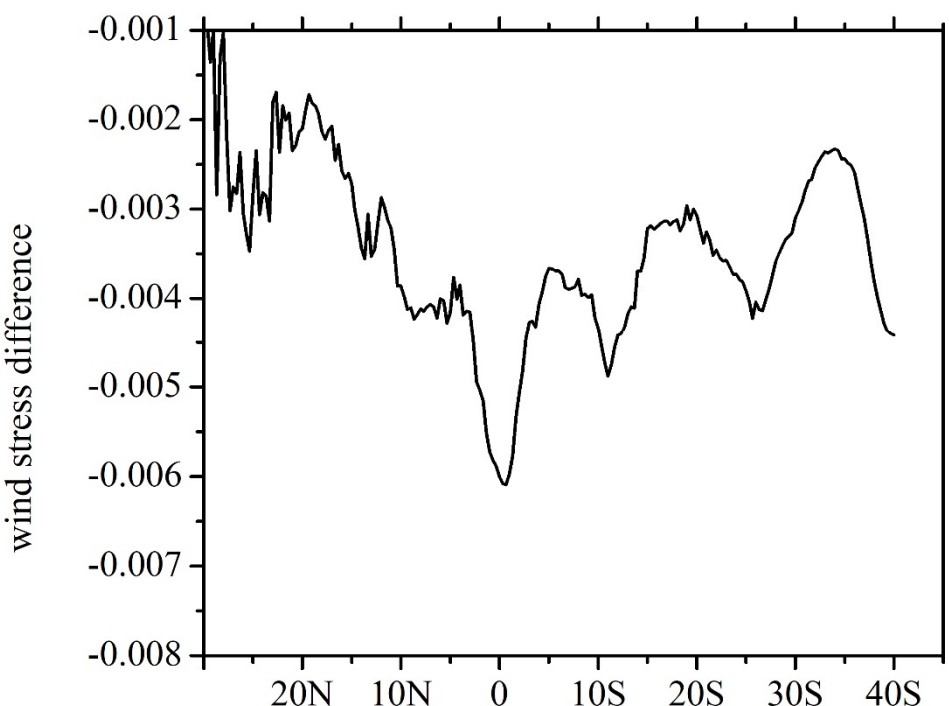

**Figure 2.** Zonal average of wind stress difference ($\tau_{\text{Cur}} - \tau_{\text{no-Cur}}$; N/m$^2$) in the tropical Indian Ocean.

**Table 1.** Area averaged wind stress during 2007–2020.

| Region of Interest | $\tau_{\text{no-Cur}}$ (N/m$^2$) | $\tau_{\text{Cur}}$ (N/m$^2$) | $\tau_{\text{Cur}} - \tau_{\text{no-Cur}}$ (N/m$^2$) | % Difference |
|---|---|---|---|---|
| Basin Average | 0.06448 | 0.06073 | −0.00282 | −5.81% |
| Somali Current (43° E–64° E; 0–15° N) region | 0.06264 | 0.05665 | −0.00599 | −9.56% |
| Equatorial Jet (60° E–90° E; 2° S–2° N) region | 0.03145 | 0.02644 | −0.00501 | −15.93% |

### 3.3. Monthly and Seasonal Impact

As the currents in the Indian Ocean are highly variable and seasonally reversing in nature, the monthly mean estimations during 2007–2020 of $\tau_{no-Cur}$ and $\tau_{Cur}$ fields are estimated and the difference between the two ($\tau_{Cur} - \tau_{no-Cur}$) are shown in Figure 3. The apparent seasonal variations of the near surface currents in the Indian Ocean can be clearly seen in the monthly mean wind stress differences. Seasonal pattern of both Somali Currents and Equatorial Currents are captured in the monthly maps of the wind stress difference. The difference is prominent along the regions of stronger currents such as Somali Currents, Equatorial Jets, and Agulhas retroflection regime. In the Somali Current regime, large difference in wind stress is observed during summer monsoon months when the current speeds are maximum climatologically. Along the equator, the wind speed differences are maximum during March–April–May and November–December, during which the Equatorial Jets are stronger. The maximum wind stress difference occurs during fall inter-monsoon season when Equatorial Jets are stronger than during spring inter-monsoon season. The region south of Sri Lanka also show higher wind stress difference due to the stronger summer monsoon currents, which prevails during June–September. It is seen that wind stress pattern strongly bear the signatures of the strongest currents and its seasonality, thus corroborating the strong influence of currents in modulating the wind stress. Ref. [2] reported that the signatures of surface currents are observed in the difference between buoy and scatterometer wind data in the Pacific Ocean. The seasonal pattern of the differences between the $\tau_{Cur}$ and $\tau_{no-Cur}$ fields (Figure 4) shows that the larger deviation between the two occurs during the summer monsoon season (July–September) and the minimal deviation during the spring season (April–May). The comparison shows that significant differences exist between the wind stress fields on seasonal and monthly scales. Thus, it could be concluded that the influence of currents should be incorporate when we estimate wind stress from scatterometer observations.

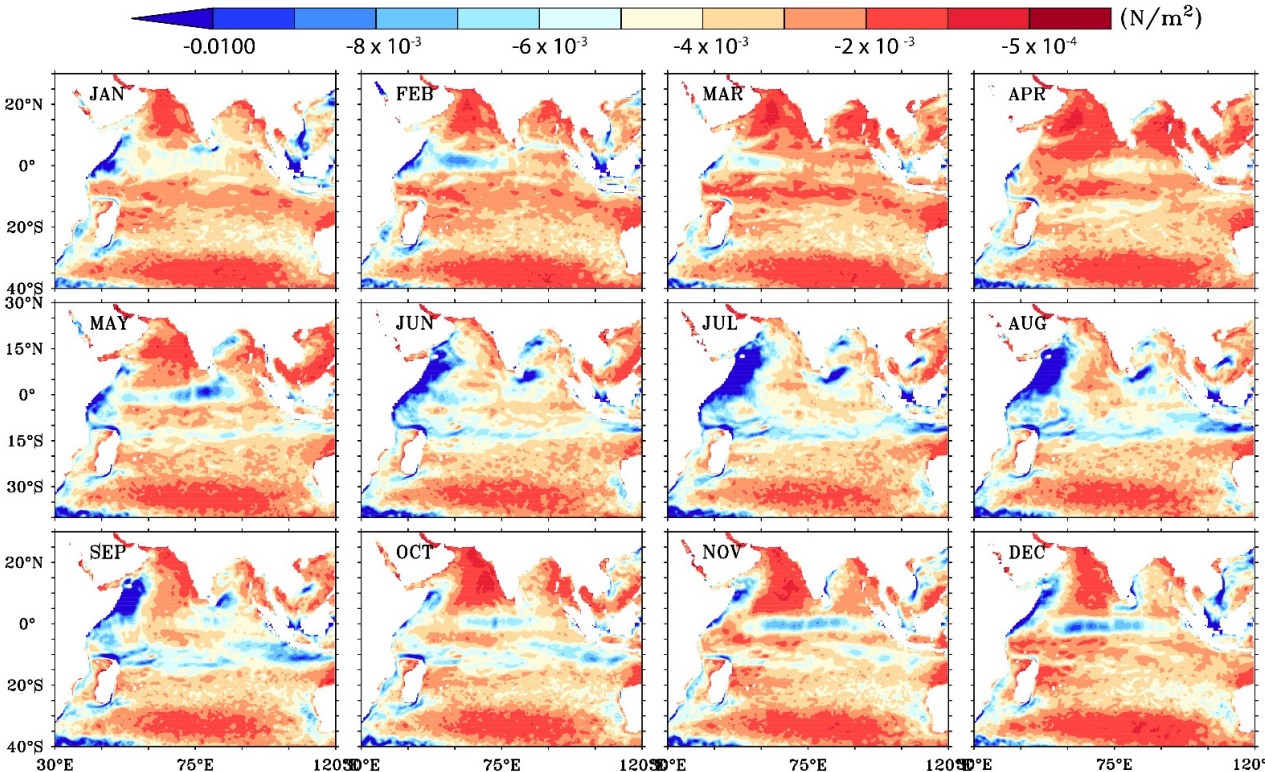

**Figure 3.** Monthly spatial variation of the difference between two wind stress estimations ($\tau_{Cur} - \tau_{no-Cur}$) averaged over 2007–2020.

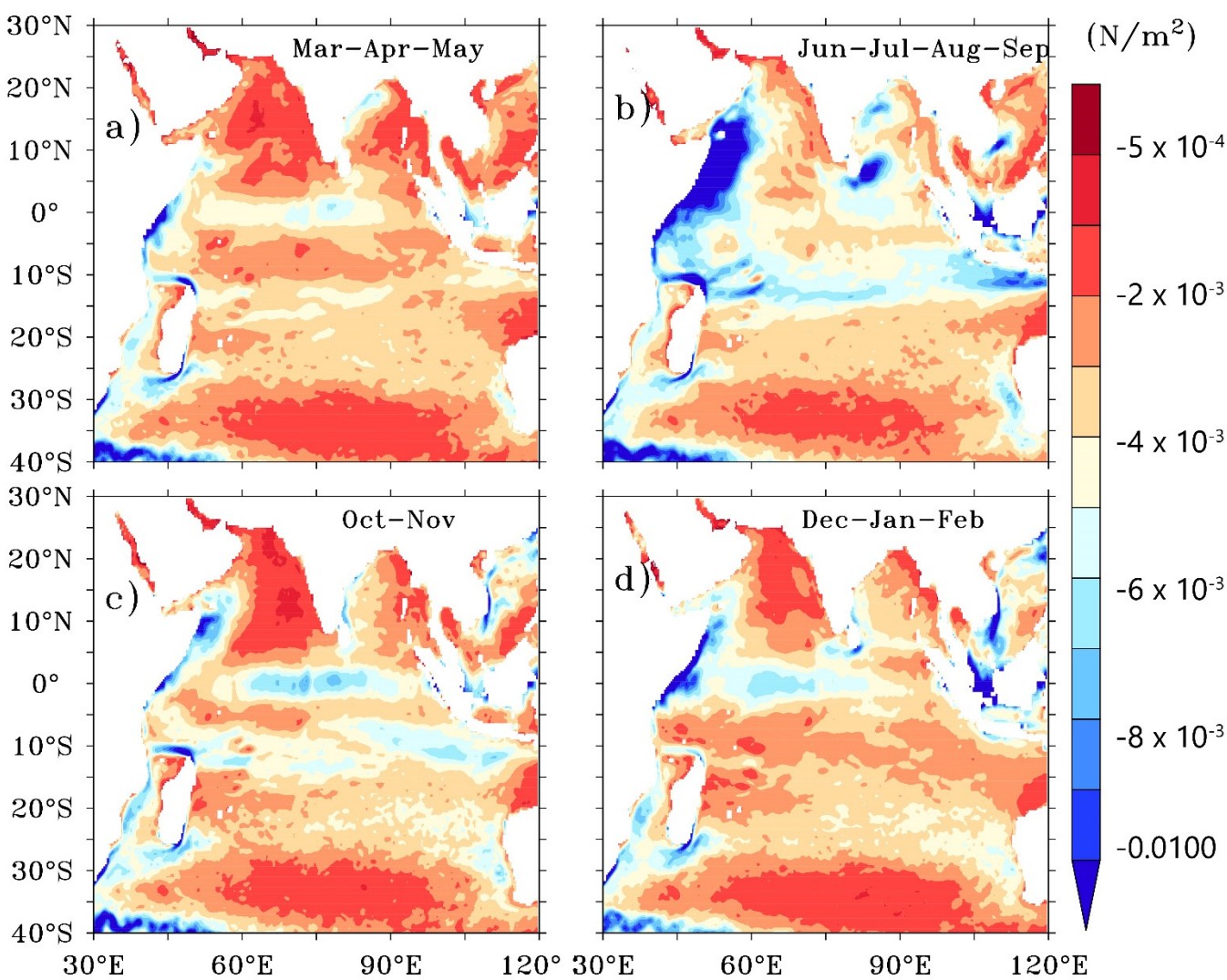

**Figure 4.** Seasonal and spatial distribution of the difference between the two wind stress estimations ($\tau_{Cur} - \tau_{no\text{-}Cur}$) averaged over 2007–2020. (**a**) Spring season (March–May), (**b**) Summer monsoon season (June–Sepetember), (**c**) Autumn season (October–November), (**d**) Winter monsoon season (December–February).

### 3.4. Sensitivity Analysis

It is seen from Figures 1–4 that the reduction in wind stress when the surface currents are included is not uniform spatially and temporally. Hence, a sensitivity analysis has been carried out in the study region for different seasons to assess the relative impact of winds and currents in the estimation of wind stress by changing the winds while keeping the currents constant and vice versa. The exercise is done to assess how the difference in wind stress estimates is sensitive to the changes in winds and current speeds. The details of the sensitivity analysis are given in the Methods section. The wind stress differences ($\tau_{Cur} - \tau_{no\text{-}Cur}$) for each sensitivity analysis averaged over the whole study region (30° E–120° E; 40° S–30° N), off Somali Coast (43° E–64° E; 0–15° N), and the equatorial region (60° E–90° E; 2° S–2° N) over the study period 2007–2020 is summarized in Table 2. The seasonal wind stress differences between the currents and with no-currents ($\tau_{Cur} - \tau_{no\text{-}Cur}$) are always negative for the three locations (Table 2), indicating that stress without incorporating currents is always larger than that with currents. The first column in the table (EXP1_W10) is by increasing the winds by 10%, the second column (EXP1_W5) is by increasing the winds by 5% without changing the currents. The third column (NOEXP)

represents the differences without changing either the currents or the winds. The fourth and the fifth column (EXP2_C5 and EXP2_C10) provide the differences by increasing the currents by 5% and 10%, respectively.

**Table 2.** Percentage change in wind stress when surface currents are accounted in the bulk wind stress equation computed from $\tau_{Cur}$ and $\tau_{no\text{-}Cur}$ for each sensitivity tests.

| Season | EXP1_W10 | EXP1_W5 | NOEXP | EXP2_C5 | EXP2_C10 |
|---|---|---|---|---|---|
| **Basin Averaged** | | | | | |
| March–April–May | −5.279% | −5.526% | −5.798% | −6% | −11.2% |
| June–September | −4.88% | −5.508% | −5.3318% | −5.59% | −9.83% |
| October–November | −5.71% | −5.98% | −6.28% | −6.562% | −15.4% |
| December–January–February | −5.896% | −6.159% | −6.611% | −6.755% | −9.8% |
| **Off Somali Coast** | | | | | |
| March–April––May | −16.11% | −16.8% | −17% | −18.43% | −19.22% |
| June–September | −12.04% | −12.55% | −13.15% | −13.76% | −14.39% |
| October–November | −15.46% | −16.1% | −16.9 | −17.6% | −18.45% |
| December–January–February | −16.48% | −17.2% | −18 | −18.85% | −19.669% |
| **Equatorial Region** | | | | | |
| March–April–May | −9.357% | −9.78% | −10.24% | −10.7% | −11.22% |
| June–September | −8.155% | −8.6% | −8.9% | −9.4% | −9.9% |
| October–November | −13.01% | −13.59% | −14.18% | −14.83% | −15.484% |
| December–January–February | −8.2% | −8.60% | −8.9% | −9.4% | −9.8% |

Compared to the NOEXP value, the differences are larger when the wind speeds are increased by 5% compared to those when increased by 10%. This indicates that the effect of currents is smaller (in a percentage sense) for higher wind speeds as the higher winds dominate over the currents on the wind stress. This is true for all seasons and the three areas studied. On the contrary, the wind stress difference increases when the current speeds are increased. These results are alarming as the ocean modelers use non-scatterometer-derived wind speeds directly in the models without applying the correction for currents. It is suggested to use at least the climatological current speeds to adjust non-scatterometer-derived wind stress. The impact of currents is highest during December–May in the Somali region when the currents speeds are low.

### 3.5. Impact on Derived Parameters

Since the ocean surface current can affect wind stress as shown in the above section, the processes that are based on wind stress also will be affected. The resultant change in the wind stress also tend to modify the wind stress derivative fields such as wind power input, Ekman currents, upwelling velocity, and wind stress curl [15,18]. Hence, we tried to assess the impact of currents on these three major parameters, i.e., wind stress curl, Ekman Currents, and wind power input into the ocean.

The curl of the wind stress is computed using the equation,

$$curl = \left( \frac{\partial \tau_y}{\partial x} - \frac{\partial \tau_x}{\partial y} \right) \tag{3}$$

where $x$ and $y$ are eastward and northward coordinates and $\tau_x$ and $\tau_y$ are the corresponding components of the wind stress. Wind stress curls are estimated using $\tau_{Cur}$ and $\tau_{no\text{-}Cur}$ and the difference between the two is illustrated in Figure 5 on an annual basis. On a large scale, the wind stress curl field over much of the Indian Ocean is very similar in $\tau_{Cur}$ and $\tau_{no\text{-}Cur}$ estimations. The inclusion of surface current feedback modulates the wind stress curl more along the equatorial Indian Ocean, Somali region, a coastal region along with the Southwest Arabian Sea, etc. The largest difference in the wind stress curl is exhibited along the equatorial Indian Ocean and the Somali region, which are the permanent upwelling systems in the Indian Ocean [23,24]. A maximum positive difference is present in the Somali and the western Equatorial Indian Ocean, while the least was present in the central Equatorial Indian Ocean.

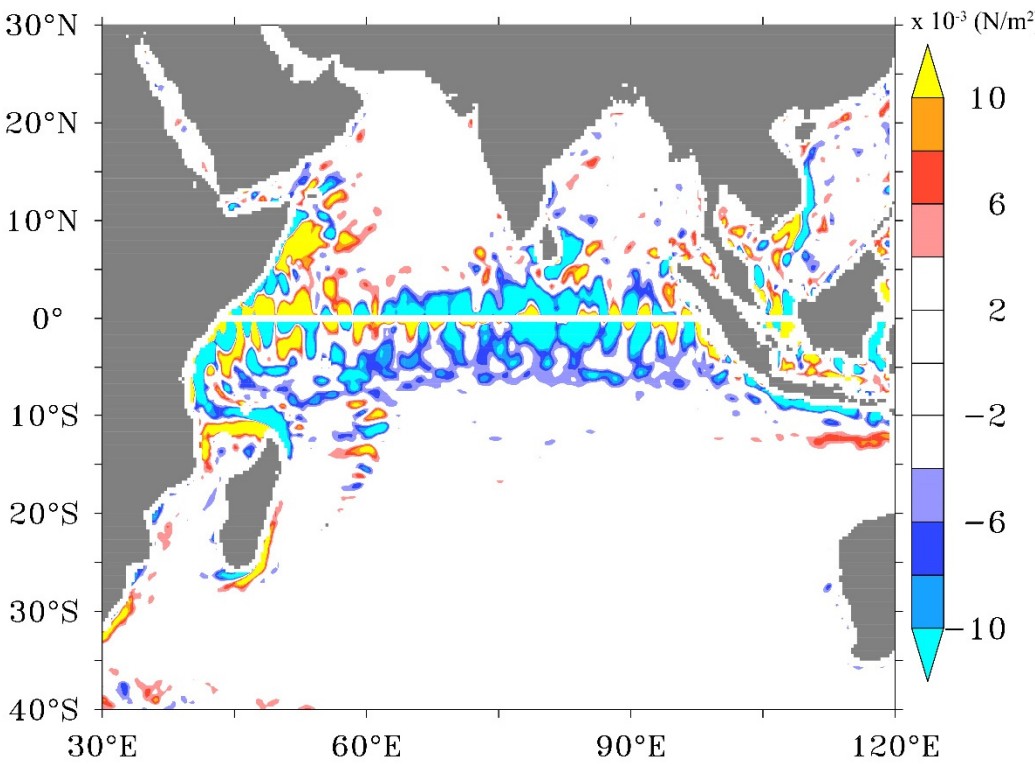

**Figure 5.** Average annual difference between the wind stress curl estimated with and without the inclusion of surface currents ($\tau_{Cur} - \tau_{no\text{-}Cur}$) during 2007–2020.

Ekman current is another important field that is dependent on wind stress. Changes in the wind stress and its curl can alter Ekman currents. To show this, we have calculated Ekman currents, $U_e$ with and without surface current using the equation

$$U_e = \frac{\tau}{\rho(A|f|)^{1/2}} \tag{4}$$

where $\rho$ is the density of sea water, $A$ is the eddy viscosity ($10^{-2}$ m$^2$/s), $f$ is the coriolis parameter, and $\tau$ is the wind stress computed without and with the inclusion of surface currents using Equations (1) and (2), respectively. The difference between the two estimations of Ekman currents is shown in Figure 6. Ekman currents are reduced along major portions of the region of study after incorporating the current effect. The weakening of Ekman currents can be explained due to the dampening of the wind stresses when currents are included. The reduced wind stress reduces the momentum transfer across the ocean surface and hence Ekman currents dampen. The change in the Ekman Currents induced by surface currents is dominant along the tropical Indian Ocean. The maximum reduction of Ekman Current occurs along the equatorial region. Minimal impact is seen in the region south of 10° S, except for a few localized regions like Agulhas currents. Similarly, little change is observed in the Arabian Sea and the western Bay of Bengal. Reduction is also visible along the eastern coast of India. Basin average percent reduction of Ekman currents by the inclusion of surface currents is ~−7% (Table 3).

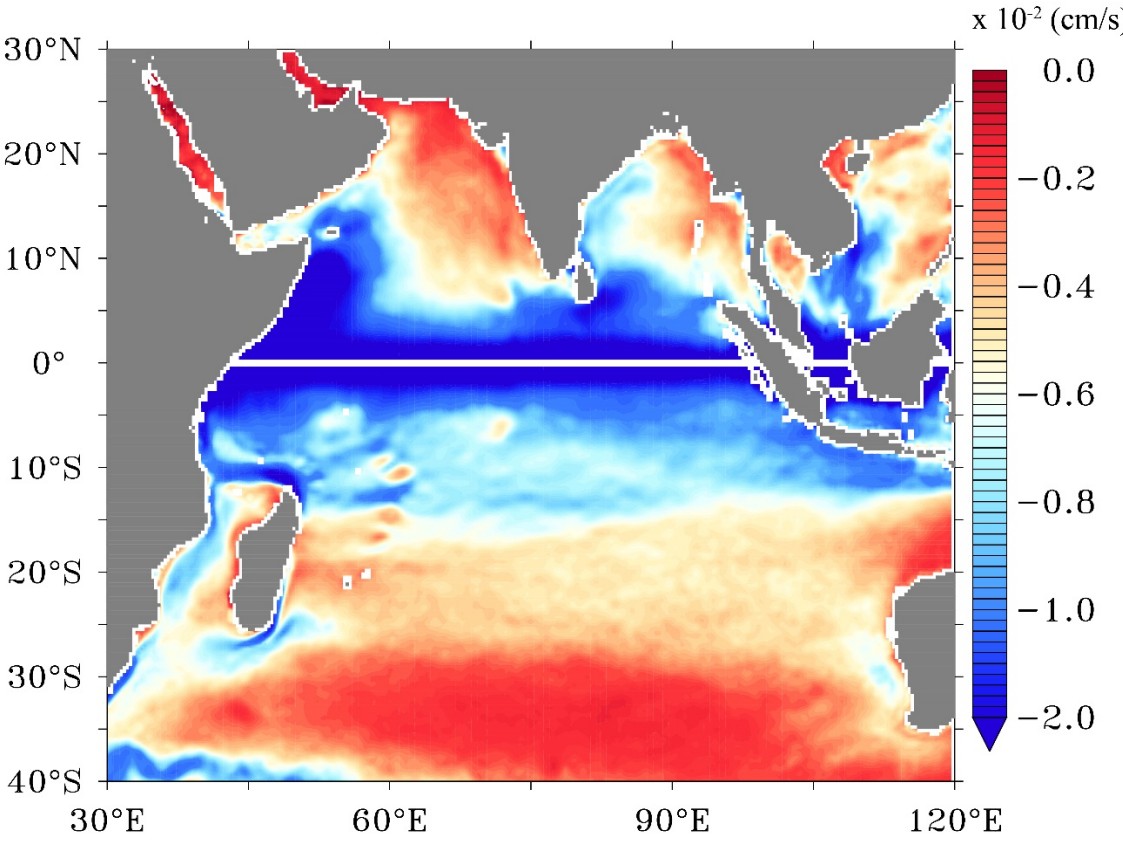

**Figure 6.** Average difference between the Ekman currents (cm/s) computed with and without the inclusion of currents in the wind stress during 2007–2020.

**Table 3.** Basin wide area averages of Ekman currents and wind power input.

| Parameter | Without Current | With Current | % Difference |
|---|---|---|---|
| Ekman Currents (m/s) | 0.1130 | 0.1050 | −5.46 |
| Wind Power input (W/m$^2$) | 0.01110 | 0.00998 | −10.0631 |

The ageostrophic wind power input into the ocean is estimated with and without the current inclusion using the equation

$$P = \tau * U_e \tag{5}$$

where $\tau$ is the wind stress without and with the inclusion of surface currents computed using (1) and (2), respectively, and $U_e$ is the surface Ekman Current. The difference between the two wind powers is shown in Figure 7. The wind power input reduction is negligible throughout the Indian Ocean with an exception over the regions off the Somali coast and a narrow band along the equator. An average basin-wide reduction of −10% is observed by the inclusion of surface currents in the wind stress estimation. [11] have reported a wind power reduction of ~25% in the North Pacific Ocean.

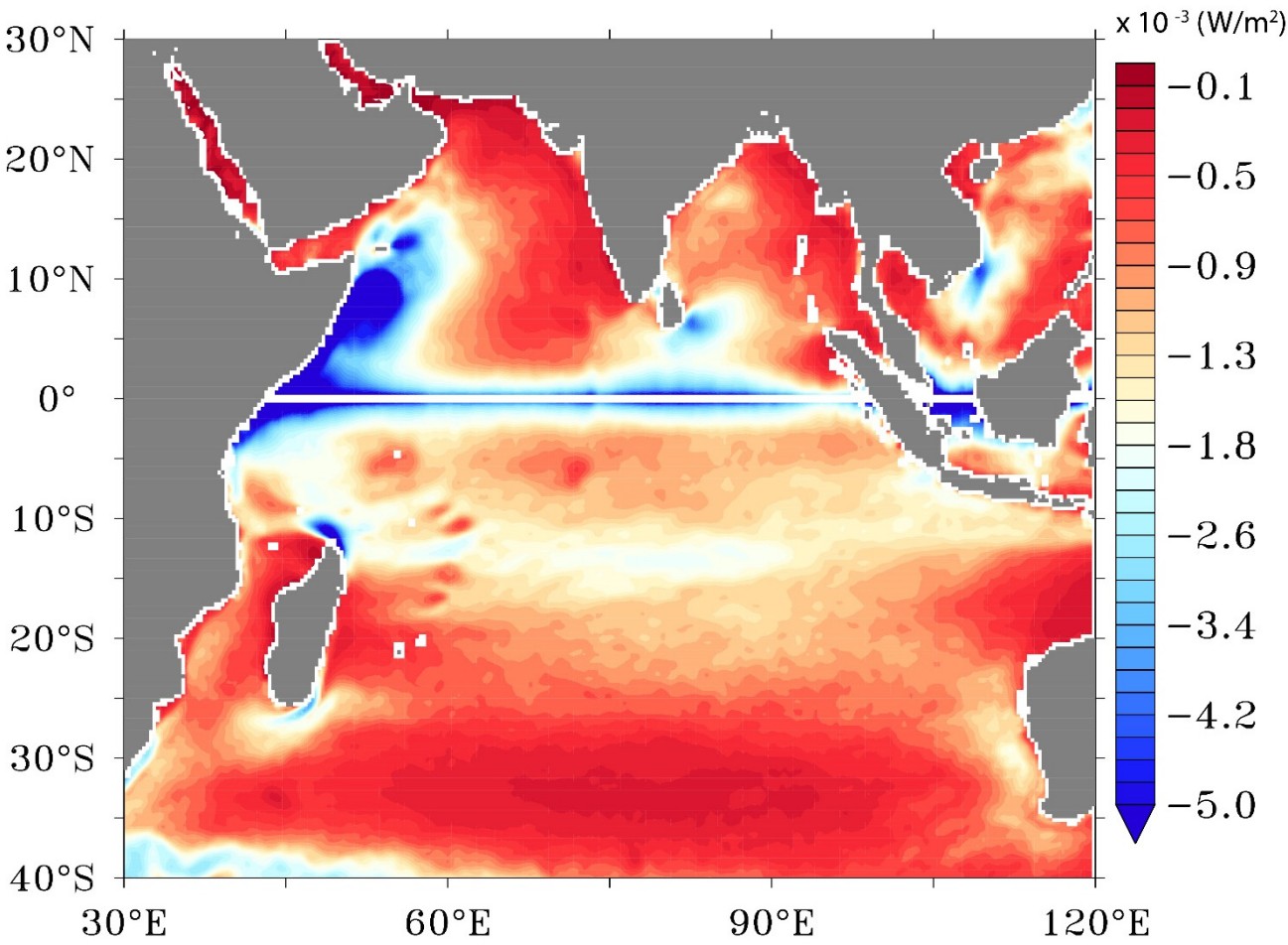

**Figure 7.** The average difference between ageostrophic wind power ($\times 10^{-3}$ W/m$^2$) estimated with and without the surface currents during 2007–2020.

The inclusion of the current speed in the wind stress bulk formula resulted in the modulation of the wind stress curl, Ekman Currents, and wind power input to the ocean. However, regional differences exist in the variation between each of these parameters.

## 4. Conclusions

In this work, we have assessed the impact of including surface current speeds in estimating wind stress using the bulk formula. Using thirteen years of satellite measurements of wind speeds and surface current observations, we find that currents can have a significant impact on wind stress estimation. A basin-wide reduction in the wind stress is observed when surface currents are included in the wind stress equations. While the basin averaged net wind stress reduction accounts for −5.8%, a relatively notable reduction of wind stress is observed in the regions off the Somali coast (−9.56%) and equatorial region (−15.93%). Sensitivity analysis has been carried out for the study region for different seasons to assess the relative impact of winds and currents in the estimation of wind stress by changing the winds, while keeping the currents constants and vice versa. The impact of currents is larger for lower wind speeds compared to the higher wind speeds, indicating that the effect of currents is smaller for higher wind speeds as the higher winds dominate over the currents on the wind stress. This is true for all seasons and the three areas studied. On the contrary, the wind stress difference is prominent for higher current speeds.

We also studied the effect of the inclusion of surface currents in the Ekman Currents, wind power input, and wind stress curl fields. The results show that the fields that are wind stress-dependent also vary, the regional differences in the variability vary for each parameter assessed. The damping effect of surface currents is strongest along the equatorial region. In general, this work highlights the importance of the inclusion of surface currents in wind stress estimation. Uncertainties in the wind stress estimations are quite large at regional levels and hence have important implications in the air–sea interaction. This implies that a better representation of wind stress by using the scatterometer-derived stress or information on currents rather than uncoupled model wind stress should for forcing ocean/climatic models for accurate air–sea interaction studies.

**Author Contributions:** Conceptualization, N.C. and M.M.A.; Methodology, N.C.; Software, N.C.; Validation, N.C.; Formal analysis, N.C.; Investigation, N.C. and M.M.A.; Resources, N.C.; Data curation, N.C.; Writing—original draft preparation, N.C.; Writing—review and editing, N.C., M.M.A. and M.A.B.; Visualization, N.C.; Supervision, M.M.A.; Project administration, N.C.; Funding acquisition, M.A.B. All authors have read and agreed to the published version of the manuscript.

**Funding:** MAB's contributions were funded in large part by NASA Physical Oceanography via the Jet Propulsion Laboratory (Contract #1419699) and funded in part by the Global Ocean Monitoring and Observing Program (Fund #100007298), National Oceanic and Atmospheric Administration, U.S. Department of Commerce through the Northern Gulf of Mexico Institute (NGI grant number 20-NGI3-106).

**Data Availability Statement:** All the data used in this study are from the public domains as mentioned in the acknowledgments section.

**Acknowledgments:** NC greatly acknowledges the support and the encouragement provided by General Manager, RRSC-east, and the Director, National Remote Sensing Centre. MMA acknowledges the support and encouragement provided at COAPS, FSU, and APSDMA. The ASCAT wind data used in this work is obtained from the Asia Pacific Data research Centre (APDRC; http://apdrc.soest.hawaii.edu/datadoc/ascat.php) (accessed on 3 September 2021). The OSCAR surface current data is downloaded from http://apdrc.soest.hawaii.edu/las/v6/dataset?catitem=2845 (accessed on 3 Sepetember 2021). The figures are generated in Ferret and Origin. The authors thank the three anonymous reviewers and the Academic Editor for their critical comments and suggestions due to which the quality of the manuscript has improved.

**Conflicts of Interest:** The authors declare no conflict of interest.

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
