# Peer review of "Impact of Ocean Currents on Wind Stress in the Tropical Indian Ocean"

_remotesensing, doi:10.3390/rs14071547_

Round 1

Reviewer 1 Report

I am generally happy with the improvements in the manuscript. Two comments still remain however and I think they should be addressed before final publication.

(1) I see that the authors have improved the colour palette in Figure 3, 4, 6, and 7 as suggested, but not Figure 1 which still has the same colour palette(which is unclear in my vision). Did you try a blue-to-red scale for Figure 1 as well? Please elaborate.

(2) It appears that the sentence ‘The ASCAT surface winds are re-gridded to match the spatial and temporal resolution of OSCAR surface currents for computing the wind stress’ is repeated twice in the text (lines 137-140).

Author Response

Replies to the comments of the Reviewer-1:

We thank the reviewer for his critical observations and suggestions to the mistakes.

(1) I see that the authors have improved the colour palette in Figure 3, 4, 6, and 7 as suggested, but not Figure 1 which still has the same colour palette(which is unclear in my vision). Did you try a blue-to-red scale for Figure 1 as well? Please elaborate.

Reply: We apologize for not changing Figure-1. Now, we changed it as suggested. Thank you for the observations.

(2) It appears that the sentence ‘The ASCAT surface winds are re-gridded to match the spatial and temporal resolution of OSCAR surface currents for computing the wind stress’ is repeated twice in the text (lines 137-140).

Reply: Again, sorry for the mistake. We removed the repeated sentence.

Reviewer 2 Report

In this revision, the authors did not properly address the major concerns raised in my previous review. A major revision or rejection is recommended.

1) It is understandable that the GMFs were developed based on the empirical relationship of the surface backscatter and in situ winds (without incorporating currents), therefore the derived satellite winds should be close to the winds alone but not the relative winds. However, Kelly et al. (2001) and others have clearly reported the characters of surface currents in the scatterometer winds when comparing them to the in situ winds. The authors need to explain why their scatterometer wind products in the tropical Indian Ocean do not include the surface current influence, or the limitations of Kelly et al. (2001) study in the tropical Pacific. If this can not be done, then an analysis parallel to Kelly et al. (2001) in the tropical Indian Ocean is needed here to demonstrate the missing ocean currents in their scatterometer winds before applying the ocean current correction in wind stress estimate.

2) Wind stress does have direction, the negative zonal wind stress means westward wind stress. In the anomaly sense, negative zonal stress anomaly means more westward stress or less eastward stress. In Uw-Uo, when Uo is opposing Uw, Uw-Uo ends up with a larger magnitude than Uw. It seems that the authors did not take the direction into consideration. 

Author Response

Replies to the comments of Reviewer-2:

We thank the reviewer for the suggestions due to which the quality of the manuscript has improved.  

Comment-1:

It is understandable that the GMFs were developed based on the empirical relationship of the surface backscatter and in situ winds (without incorporating currents), therefore the derived satellite winds should be close to the winds alone but not the relative winds. However, Kelly et al. (2001) and others have clearly reported the characters of surface currents in the scatterometer winds when comparing them to the in situ winds. The authors need to explain why their scatterometer wind products in the tropical Indian Ocean do not include the surface current influence, or the limitations of Kelly et al. (2001) study in the tropical Pacific. If this can not be done, then an analysis parallel to Kelly et al. (2001) in the tropical Indian Ocean is needed here to demonstrate the missing ocean currents in their scatterometer winds before applying the ocean current correction in wind stress estimate.

Reply:

We now agree that scatterometers measure surface relative winds.  We have modified the text to address the impacts of calculating stress from Earth-relative winds rather than satellite relative winds. The Earth relative winds are determined by the vector addition of scatterometer winds and currents.

Comment 2:

Wind stress does have direction, the negative zonal wind stress means westward wind stress. In the anomaly sense, negative zonal stress anomaly means more westward stress or less eastward stress. In Uw-Uo, when Uo is opposing Uw, Uw-Uo ends up with a larger magnitude than Uw. It seems that the authors did not take the direction into consideration. 

Reply:

Stress direction is dealt with as vector components in the calculation of Ekman upwelling. Our stress direction is equal to the surface relative wind direction.

Reviewer 3 Report

The revised manuscript has been improved a lot and is recommended to be accepted for publication.

Author Response

Reply to the comment of Reviewer-3:

The revised manuscript has been improved a lot and is recommended to be accepted for publication.

Thank you very much for accepting the manuscript.

This manuscript is a resubmission of an earlier submission. The following is a list of the peer review reports and author responses from that submission.

Round 1

Reviewer 1 Report

Bulk surface flux calculations rely on relative wind (near surface wind relative to surface ocean current).  This study attempts to assess the impact of ocean surface current (if neglected) on wind stress estimation using satellite ASCAT scatterometer wind product, with particular attention to the Indian Ocean. It also tries to estimate the impact of ocean current on wind stress curl and wind work, if neglected. I recommend this manuscript to be rejected. The authors are encouraged to resubmit it after the following concerns are addressed. 

Major concern:

Satellite scatterometers measure more surface wind stress (relative wind) than wind, as reported by Kelly et al. (2001) and others. This is stated in the Introduction (Line 41 and various other places) of this manuscript. Backscatters measured by scatterometers are directly related to wind stress and relative wind. Kelly et al. and others claimed this as the advantage of satellite winds over in situ measurements (which often do not measure surface currents). Some of the satellite wind products were tuned to winds from in situ measurements. This adds some confusion. But clearly, Kelly et al. and others did find out that satellite wind products are more representative of the relative winds. 

Shouldn’t all of these suggest that wind stress estimates using scatterometer winds won’t need to be referenced to the surface current anymore?  

Specific comments:

1)    Line 92-93: Change  “ … Somali currents … flows northwards … southward …” to “ … Somali current … flows northeastwards … southwestwards …”
2)    Line 115: Any reference for the ASCAT winds?
3)    Line 157 and Fig.1: It is hard to believe that ocean currents contribution to the wind stress estimate are all negative in the entire Indian Ocean. When ocean currents happen to be in the opposite direction to the wind, adding ocean currents to the bulk stress calculation would increase the wind stress. It is entirely possible that the currents in the Indian Ocean are all in the direction of surface winds. But please show the mean wind vectors in Fig1a. and ocean currents in Fig.1b.
4)    Line 158-159: Please make sure the following statement is correct: “that stress without currents is more than the stress without current”.
5)    Line 164-166: Change “Since the currents are more in this region, the difference is also more.” to “ Since the currents are STRONGER in this region, the difference is also LARGER.”
6)    What is value of Cd in equation (2)? How is it determined?
7)    Line 231-233: See Major Concern.
8)    Line 266-268, equation (4): Is this the vertically averaged Ekman current? Please double check the equation? Shouldn’t there be a “2” in (2Af)^(1/2)? 
9)    What is the value of A used for calculating Ekman velocities?
10)    Shouldn’t the surface Ekman velocity, instead of the vertically averaged Ekman velocity, be used in equation (5) to calculate the wind work?  Please justify.

Reviewer 2 Report

The authors present the results of a study where they investigate the impact of surface currents on the calculation of wind stress from scatterometer observations in the Indian Ocean.

I am generally happy with the paper and I think that it can be published after consideration of the comments below. My suggestions may be useful to further improve the manuscript.

First a general remark. I would strongly suggest to change the colour palette in Figures 1, 3, 4, 6 and 7. The current palette goes from a dark colour (purple) via a light colour (yellow) back to a dark colour (red/brown) which is confusing in my opinion and makes it difficult to distinguish between lower and higher values on the maps. I suggest to try a single colour palette, e.g. going from white to dark red or going from white to dark grey.

Page 1, line 34-36: Please explain the meaning of the brackets ‘()’ and bars ‘||’ in equation 1.

Page 3, line 113: ‘European Organization’ should be ‘European Organisation for the Exploitation of Meteorological Satellites (EUMETSAT)’ I guess?

Page 3, line 114: I suggest to mention the source of the ASCAT data here for clarity: Asia Pacific Data research Centre (APDRC). I have seen that it is named in the acknowledgements at the end of the manuscript but it is better to mention the data provider here as well like is done for the OSCAR currents.

Page 3, lines 127-130: Here the authors explain how they compute the stresses with and without ocean currents. It is not clear however how they perform the temporal averaging. Are the ASCAT winds averaged over 5 days (the temporal resolution of the currents) before computing stress and are the obtained 5-day stresses then time averaged over a month, season or year? Or are the winds and stresses first averaged separately to a month/season/year and is the stress computed based on time averaged winds and stresses? I assume that both methods may produce significantly different results. Please elaborate on this in the text and justify the chosen averaging strategy.

Page 5, lines 164-166: please replace ‘more’ by ‘larger’ twice.

Page 7, line 221: ‘is always more than’ -> ‘is always larger than’.

Page 7, line 227: ‘the differences are more when’ -> ‘the differences are larger when’.

Page 11, line 313: ‘is more for lower wind speeds’ -> ‘is larger for lower wind speeds’.

Please also check the use of ‘less’ throughout the text. In my understanding it should be replaced by ‘smaller’ in some cases.

Reviewer 3 Report

This manuscript present the impact of ocean currents on wind stress in the Tropical Indian Ocean. However, there are two main issues which are not appropriate.

One issue is that the OSCAR current is not the current that related to the wind stress. It's derived using satellite sea surface height and a quasi-steady geostrophic model. 

The other issue is that the surface current modify the surface roughness which changes the interaction between the wind and the sea surface. The relationship should not be simply considered as ??−??.

I strongly recommend the authors reconsider the method and the surface current data used.